# Effects of Cultivar Factors on Fermentation Characteristics and Volatile Organic Components of Strawberry Wine

**DOI:** 10.3390/foods13182874

**Published:** 2024-09-11

**Authors:** Wei Lan, Mei Zhang, Xinyu Xie, Ruilong Li, Wei Cheng, Tingting Ma, Yibin Zhou

**Affiliations:** 1College of Food and Nutrition, Anhui Agricultural University, Hefei 230036, China; lanwei@fynu.edu.cn; 2Anhui Engineering Research Center for Functional Fruit Drink and Ecological Fermentation, Fuyang 236037, China; 22211310@stu.fynu.edu.cn (M.Z.); xiexinyu710@163.com (X.X.); pennylrl@163.com (R.L.); 13805585071@163.com (W.C.); 3School of Biology and Food Engineering, Fuyang Normal University, Fuyang 236037, China; 4Shaanxi Provincial Key Laboratory of Viti-Viniculture, Shaanxi Engineering Research Center for Viti-Viniculture, College of Food Science and Engineering, Northwest A&F University, Yangling 712100, China; matingting@nwafu.edu.cn

**Keywords:** strawberry wine, cultivar, physicochemical indicators, volatile organic components, fermentation

## Abstract

Strawberry wine production is a considerable approach to solve the problem of the Chinese concentrated harvesting period and the short shelf life of strawberries, but the appropriative strawberry cultivars for fermentation are still undecided. In this study, the strawberry juice and wines of four typical strawberry cultivars named Akihime (ZJ), Sweet Charlie (TCL), Snow White (BX), and Tongzhougongzhu (TZ) were thoroughly characterized for their physicochemical indicators, bioactive compounds, and volatile organic components (VOCs) to determine the optimal strawberry cultivars for winemaking. The results showed that there were significant differences in the total sugar content, pH, total acid, and other physicochemical indexes in the strawberry juice of different cultivars, which further affected the physicochemical indexes of fermented strawberry wine. Moreover, the content of polyphenols, total flavonoids, vitamin C, and color varied among the four strawberry cultivars. A total of 42 VOCs were detected in the strawberry juice and wines using headspace solid-phase microextraction coupled with gas chromatography–mass spectrometry (HS-SPME-GC-MS), and 3-methyl-1-butanol, linalool, trans-2-pinanol, hexanoic acid, and hexanoic acid ethyl ester were the differential VOCs to identify the strawberry wine samples of different cultivars. Overall, strawberry cultivar ZJ had a relatively high VOC and bioactive compound content, indicating that it is the most suitable cultivar for strawberry wine fermentation. In addition to determining the relatively superior fermentation characteristics of cultivar ZJ, the results may provide a theoretical basis for the raw material quality control and quality improvement of strawberry wine.

## 1. Introduction

Strawberry (*Fragaria × ananassa*) is one of the most indispensable berries in the world, with a considerable production of approximately 9 million tons in 2019 [1]. China has the largest strawberry planting area and output worldwide, creating more than USD 5 billion every year [2]. Strawberry fruits have bright colors and a unique aroma, and contain various nutrients such as organic acids, amino acids, vitamins, and phenolic constituents, which are deeply appreciated by consumers [3]. In addition, the benefits and biological functions of strawberry in terms of antioxidant, anti-inflammatory, and anti-cancer activity have been verified by relevant in vitro and in vivo studies [3,4,5]. Although greenhouse technology allows strawberries to mature all year round, strawberries in China are mainly harvested in late spring. Due to the susceptibility to damage, strawberries are perishable, and this, coupled with a short shelf life, results in a large amount of strawberry fruits being wasted every year [6]. Strawberry wine production is an considerable approach to extend the shelf life and increase the added value of strawberries, while the fermentation process effectively preserves the nutrients in the strawberry fruit [7].

Fermented fruit wines have gained great interest in recent years. Honey peach [8], pineapple [9], kiwifruit [10], melon [11], and red dragon fruit [12] can be fermented into fruit wine, and the multiformity of raw materials contributes to the diversity of sensory and flavor characteristics in fruit wine [13]. Moreover, the vitamins, polysaccharides, and polyphenolic compounds in fruit wine have anti-oxidation effects, lowering blood lipids and preventing cardiovascular diseases, which caters to society’s demand for healthy drinking [14,15]. Aroma is a crucial factor that determines the preference and value of fruit wine [16]. Factors that determine the aroma of fruit wine include the quality of fruit raw materials, fermentation strains, fermentation protocol, and aging technologies, among which the quality of fruit raw materials is the key element considering that fermented fruit wine is rarely aged in oak barrels [13]. And the quality of fermented fruit raw materials is influenced by various factors such as cultivar, geographical climate, agronomic management, and post-harvest treatment [17]. China has a wide range of strawberry cultivation areas, with over 200 strawberry cultivars. The interaction between geographical climate and cultivars leads to significant differences in physicochemical properties of strawberry fruit, but the instability of raw material quality has led to limited sales of strawberry wine on the Chinese market [7].

Šamec revealed differences in physicochemical properties, antioxidant properties, and phytochemical properties among four strawberry cultivars, Albion, Monterey, Capri, and Murano, through principal component analysis (PCA). The results indicated that cultivar factors had a greater impact on strawberry fruit quality than the harvesting date. Cultivar Monterey had the largest quantity of phytochemical contents and consequently the strongest antioxidant activity, while Albion had the highest content of soluble solids, titratable acidity, and ascorbic acid [18]. Nowicka disclosed the qualitative and quantitative composition of polyphenols in 80 cultivars of strawberry fruits employing ultra-performance liquid chromatography/quadrupole time-of-flight mass spectrometry (UPLC-QTOF-MS/MS), and screened out the strawberry cultivars with high tannins and antioxidant properties [19]. Wang conducted non-targeted metabolomic analysis, finding 142 metabolites in ten strawberry cultivars; these compounds were divided into six categories, and 72 biomarkers among different cultivars were identified [20]. In addition, Sheng detected the aroma-related VOCs in the strawberry fruit of 16 cultivars, and the characteristic aroma was identified as nine esters, six aldehydes, and one alcohol [21]. Whereas the fermentation quality of fruit depends on more than just its chemical and aroma composition, the fermentation characteristics of different cultivars of strawberries remain elusive [22].

Cultivar selection is a prerequisite for the quality control of fruit wine raw materials; however, knowledge regarding the fermentation quality of different cultivars of strawberries is still limited [15]. Therefore, this paper aims to reveal the fermentation characteristics of four typical strawberry cultivars which are widely cultivated. By monitoring the fermentation process and detecting the physicochemical indicators, bioactive compounds, and VOCs in strawberry juice and wines, the brewing characteristics of these cultivars are efficaciously evaluated.

## 2. Materials and Methods

### 2.1. Strawberry Cultivars and Yeast Strains

This study included four cultivars of strawberries, named Akihime (ZJ), Sweet Charlie (TCL), Snow White (BX), and Tongzhougongzhu (TZ), which were collected at the strawberry planting base in Wenji Town (N: 33°03′22.50″, E: 115°38′43.60″), Fuyang City, Anhui Province, between March and April 2022. These sampled strawberry fruits were stored at −20 °C immediately after screening until processing. Commercial yeast strain Lalvin Rhône 2323 (SC, Lallemand Inc., Montreal, QC, Canada) was applied to fermentation.

### 2.2. Preparation of Strawberry Juice and Wine Fermentation

Five kg of frozen strawberry fruits were thawed at 25 °C for 2 h, and then crushed using a juice press. SO_2_ as metabisulphite was added to the acquired strawberry juice at 30 mg/Kg, followed by 300 mg of pectinase Lallzyme EX-V (30 mg/L), and incubated for 2 h at 40 °C [23]. The enzyme-treated strawberry juice was centrifuged at 8000× *g* for 10 min at 4 °C and Brix was adjusted to 23° with sucrose [10]. The adjusted strawberry juice was pasteurized at 97 °C for 30 s using a hot water bath, controlled with a thermometer, and then immediately cooled to room temperature with an ice bath. For each fermentation, 300 mL of pasteurized strawberry juice was immediately transferred to a 500 mL sterile jar.

Commercial yeast strain Lalvin Rhône 2323 was activated for 12 h at 28 °C in liquid YPD medium and diluted strawberry juice, respectively, ensuring the colony concentration above 106 CFU/mL, and 1% *v*/*v* strawberry juice preculture was inoculated to each flask to start fermentation. All fermentations were performed in triplicate and the fermentation temperature was controlled at 25 °C (±2 °C) [24]. In the first three days, the flasks were sealed with sterile sealing film. After fermentation started, plastic film was added to the sealing film to maintain an anaerobic environment.

Except for the first day, Brix was measured every two days to monitor the fermentation progress. If there was no change in three consecutive measurements, it was considered that the strawberry wine sample had completed the alcoholic fermentation. Samples of 5 mL each were collected at 2 d, 4 d, 6 d, 8 d, and 10 d for oenological parameter analysis such as reducing sugar, alcohol, and glycerol. Moreover, 50 mL juice and fermented wine (10 d) was sampled for physicochemical indexes such as pH and total acid, functional composition, chromatic value, and VOC analysis [25].

### 2.3. Physicochemical Index Determination

#### 2.3.1. Oenological Parameters Analysis

The ◦Brix and pH values were measured using a refractometer (ATGO, Tokyo, Japan) and a pH meter (FE28, Mettler Toledo, Greifensee, Switzerland). And total acid was analyzed with the relevant chemical kits using the full-automatic analyzer (Y15, BioSystem, Barcelona, Spain).

The glucose, fructose, and ethanol content were determined by high-performance liquid chromatography (HPLC) with a LC-16system (Shimadzu, Kyoto, Japan) using an RID-20A RI detector and a Bio-Rad Aminex HPX- 87H resin-based column (300 × 7.8 mm), eluted with 5 mM H_2_SO_4_ at 55 °C and 0.5 mL/min [26].

#### 2.3.2. Nutritional Components Analysis

The content of total polyphenols, total flavonoids, and vitamin C was analyzed with the relevant chemical kits applying the full-automatic analyzer (Y15, BioSystem, Spain).

#### 2.3.3. Chromatic Value Analysis

The chromatic difference was measured using a chrominometer (CR-400, Konica Minolta, Inc., Tokyo, Japan). The brightness index L* and color indices a* (red) and b* (yellow) for each sample were measured applying whiteboard color as standard.

### 2.4. Analysis of VOCs by HS-SPME-GC-MS

Gas chromatography–mass spectrometry (GC-MS, 8860-5977C, Agilent, Palo Alto, CA, USA) and a DB-Wax column were utilized to determine VOCs in strawberry juice and wine samples according to the method in [27] with minor modifications. Automatic HS-SPME was performed with a Smart SPME Arrow, DVB/carbon WR/PDMS 1.10 mm × 120 µm SPME fiber (Agilent, CA, USA) on a CTC CombiPAL autosampler (CTC Analytics, Zwingen, Switzerland). Briefly, 5 mL of sample and 10 μL of internal standard (4-methyl-2-pentanol, 50 mg/L) were placed in a 20 mL headspace vial.

All volatile compounds were identified by comparing the mass spectrometry data with the NIST20 library. The quantitative analysis of identified volatile compounds was performed by comparing the peak areas with the internal standard.

### 2.5. Statistical Analysis

All determinations were performed in triplicate and displayed with mean value and standard deviation. Statistical analysis was performed using SPSS 21.0 (IBM, Armonk, NY, USA). The statistical significance of the data was evaluated through analysis of variance (ANOVA) using the Duncan test, and the significance level in the analysis was considered at *p* < 0.05. Unconstrained PCA was performed to analyze the difference in VOCs in strawberry juice and wine samples using SIMCA software (version 14.1) (UMETRICS, Umea, Sweden). Based on the PCA results, constrained orthogonal partial least squares discriminant analysis (OPLS-DA) was performed to distinguish the similarities and differences between the volatile compounds using R software with the DiscriMiner package (version 6.3–73).

## 3. Results and Discussion

### 3.1. Fermentation Kinetics

The fermentation kinetics analysis of strawberry juices of four cultivars is shown in Figure 1. The Brix value result showed that three strawberry cultivars (except BX) almost accomplished alcohol fermentation on the sixth day, and TZ was close to complete fermentation on the fourth day, whereas cultivar BX was not able to finish fermentation until the tenth day (Figure 1A). The results of residual sugar content and alcohol content were consistent with the Brix values. Except for cultivar BX, the other three cultivars were close to complete fermentation on the sixth day, with an alcohol content of over 10% *v*/*v*, while the fermentation process of cultivar BX was relatively slow (Figure 1B,C). And the results of glucose and fructose content elucidated that all four cultivars depleted glucose on the sixth day, and then fully utilized fructose around the tenth day (Figure 1D,E).

There are many factors determining the fermentation rate, such as the fermentation condition and yeast inoculation methods. It is a trend to improve the aroma of fruit wine and reduce the alcohol content through mixed fermentation applying non-saccharomyces cerevisiae [28]. In addition to inoculating non-saccharomyces cerevisiae for mixed strain fermentation [29], freeze-dried immobilized kefir culture [30] and high-temperature treatment [31] can also reduce the alcohol content of fruit wine; however, the effective promotion of fermentation strategies to industrial applications is still a problem to be solved [13]. For strawberry wine, further research is needed regarding fermentation condition optimization and microbial strain selection, after designating the wine-making cultivar.

### 3.2. Oenological Parameters of Strawberry Juice and Wines

Figure 2A illustrates that after alcohol fermentation, the ethanol content of all four strawberry cultivars was more than 13% *v*/*v*, and the residual sugar content was less than 4 g/L (Figure 2B), which attained the standard of dry fruit wine. Moreover, the pH value of strawberry juice varied among different cultivars; the white cultivar BX had the highest pH value of 4.03, while the cultivar TCL had the lowest pH value of 3.27 (Figure 3A). This trend was preserved in the strawberry wine, and there was also a significant difference between the pH values of strawberry wine among the different cultivars (Figure 3B). Although different cultivars of strawberry wine have achieved an alcohol content of over 12% and met the dry fruit wine standards through fermentation, the differences in strawberry juice have resulted in significant differences in the pH value and total acidity of strawberry wine, which may affect substance metabolism during the fermentation process and further characterize the quality characteristics of strawberry wine such as flavor and color [32].

### 3.3. Bioactive Compound Content of Total Polyphenols, Total Flavonoids, and Vitamin C

With the increase in people’s health awareness, fruit wine rich in bioactive substances such as polyphenols is gradually favored by the market. In addition to physicochemical and sensory quality, the content of functional factors has also become a fundamental indicator for evaluating the quality of fruit wine [33]. Consistent with the results of previous studies, the bioactive substances in strawberry juice were mainly polyphenols [19], and the polyphenol content of strawberry juice varied among different cultivars. The polyphenol content of cultivar TCL was the highest, reaching 1372 mg/L, while the contents of cultivars ZJ and BX were relatively low, at 862 mg/L and 832 mg/L, respectively.

After fermentation, the polyphenol content in strawberry wine slightly decreased. The content in cultivar TCL was still the highest at 1192 mg/L, while the content in cultivar ZJ was the lowest at 643 mg/L (Figure 4A). The content of flavonoids in strawberry juice of different cultivars also showed significant differences, with cultivar TCL having the highest content (257 mg/L) and ZJ having the lowest content (164 mg/L). After fermentation, the flavonoid content in cultivar TCL strawberry wine decreased to 172 mg/L, and that in cultivar ZJ wine decreased to 65 mg/L (Figure 4B). Unlike the results of polyphenol and flavonoid content, the strawberry juice of cultivar TZ had the highest content of vitamin C at 149 mg/L, followed by that of cultivar BX at 97 mg/L, and that of cultivar ZJ had the lowest content at only 7 mg/L. After fermentation, the content of vitamin C in cultivar TZ and BX strawberry wine still remained at or above 70 mg/L, at 70 mg/L and 74 mg/L, respectively (Figure 4C). Although the polyphenol content of strawberry fruit wine was not as high as that of cherry wine, strawberry fruit wine still possessed good antioxidant activity [34]. The strawberry fruit wine of specific cultivars is incomparable in terms of taste and health preferences in the current fruit wine market.

Through the above research, we found that there were differences in physicochemical indexes and bioactive substance contents among different cultivars of strawberries, which remained in the fermented strawberry wine.

### 3.4. Color

Color is one of the determinant factors in the sensory quality and consumer preference of fruit wine [20]. Table 1 revealed that the L* value of strawberry wine made from four cultivars demonstrated significant differences, with cultivar TZ having the highest value, showing relatively high brightness, followed by BX and ZJ, and TCL displaying the lowest L* value. Cultivar TCL had the highest a* value, followed by ZJ and BX, while TZ had the lowest a* value, indicating that cultivars TCL and ZJ were more inclined towards red tones. The b* values of the four strawberry cultivars were relatively close, and only the b* value of cultivar BX was close to 0, compared with the chrominometer whiteboard.

In addition, there were significant differences in the total color difference (E) of strawberry wine fermented from the four cultivars. The color of fruit wine is influenced by various factors such as the raw materials, fermentation microorganisms, fermentation protocol, and storage technology [35]. In addition to affecting stability and visual effects, color also affects the physicochemical properties and flavor quality of fruit wine [36]. More in-depth research is needed to explore the factors affecting the color of strawberry wine and its impact on sensory quality [37].

### 3.5. VOC Analysis of Strawberry Juice and Wines

#### 3.5.1. VOCs Analyzed by HS-SPME-GC-MS

The aroma quality is the core factor affecting the preference for fruit wine, and VOCs are paramount components of the aroma structure of fruit wine [38]. A total of 42 VOCs were detected in the strawberry juice and wine samples by HS-SPME-GC-MS (Appendix A), including 15 esters, 5 alcohols, 3 aldehyde-ketones, 7 acids, and 14 other compounds, and 38 VOCs were detected in strawberry juice and wines, respectively. While octanoic acid ethyl ester, ethyl-9-decenoate, and trans-2-pinanol were the main VOCs in strawberry juice, 3-methyl-1-butanol, octanoic acid ethyl ester, phenylethyl alcohol, hexanoic acid ethyl ester, and geranyl vinyl ether were the dominating VOCs in strawberry wine. After fermentation, the contents of alcohols, esters, and phenyl compounds in strawberry wine significantly increased, and the contents of terpenes, organic acids, and other substances in cultivar BX were relatively high (Table 2).

#### 3.5.2. Coordinate Analysis of VOCs in Strawberry Juice and Wines

Unconstrainted PCA and constrainted OPLS-DA were performed to more intuitively characterize the differences in the VOCs of different cultivars of strawberry juice and wines. As shown in Figure 5A, the combined interpretation rate of PC1 and PC2 reached 65.7%, which could effectively describe the differences in VOCs among samples. There was a significant difference in VOCs between strawberry juice and strawberry wine samples, with corresponding sample points located in the first and third quadrants, respectively. The strawberry wine sample points of cultivar BX were relatively far away from other cultivars, and the difference between the strawberry juice sample points of cultivar BX and those of other cultivars was greater, as the strawberry juice sample points of cultivar BX were located separately in the second quadrant.

The differential VOCs between strawberry juice and wine samples were α-bisabolol, hexanoic acid, 3-methyl-1-butanol, and 2-methyl-propanoic acid. The OPLS-DA analysis results were generally consistent with the PCA results (Figure 5B), with strawberry juice and wine samples located in the upper and lower quadrants, respectively. Moreover, the distance between the cultivar ZJ and TZ strawberry juice and wine samples was relatively close, indicating that their VOCs were relatively similar. VIP analysis identified 16 differential VOCs with a VIP value greater than 1 (Figure 5C); they were 3-methyl-1-butanol, linalool, trans-2-pinanol, hexanoic acid, hexanoic acid ethyl ester, and so on.

#### 3.5.3. Hierarchical Cluster Analysis of Differential VOCs in Strawberry Juice and Wines

As shown in Figure 6, the cluster heatmap analysis based on differential VOCs in strawberry juice and wine samples implied that the differences between the strawberry juice of cultivar BX and the strawberry wine of ZJ and other samples are relatively obvious. In the strawberry juice of cultivar BX, the contents of α-terpineol, linalool, α-bisabolol, 2-methyl-butanoic acid, and hexanoic acid were relatively higher; these VOCs endow the strawberry juice of cultivar BX with fruity, sweet, floral, and other aromas. On the other hand, the contents of phenylethyl alcohol, octanoic acid, 3-methyl-1-butanol acetate, and geranyl vinyl ether in the strawberry wine of cultivar ZJ, which give the wine cheesy, floral, flowery, and other aromas, were relatively higher.

Unlike wine, studies on the correlation between cultivar, origin, and fruit wine quality are still limited, especially research on the fermentation characteristics of different strawberry cultivars [13,39]. Current research mainly focuses on the influence of environmental factors and maturity stages on the composition and antioxidant activity of strawberry fruits of different cultivars [40,41,42], and the identification of cultivars and their impact on fruit quality [43]. According to the above research, it was found that among the four popular strawberry cultivars, the strawberry juice of cultivar BX was rich in terpenes and more suitable for fresh consumption. And the strawberry wine of cultivar ZJ had a relatively high content of VOCs, indicating that ZJ may be a suitable cultivar for strawberry wine brewing. Considering the large scale of strawberry cultivation in China, this study not only provides processing solutions for surplus strawberry fruits, but also helps to select strawberry varieties for planting, and there is no doubt that strawberry fruit wine has the strength to win a place in the booming fruit wine market [44]. In addition, only 42 volatile compounds were found in strawberry juice and wine samples in this study, and diverse research methods could be used to explore the aroma of strawberry wine. The comprehensive evaluation system combining GC-MS, electronic nose, electronic tongue, and sensory evaluation is a valid program to evaluate the flavor quality of strawberry wine, and the sensory evaluation based on consumer preferences is an indispensable support for strawberry wine to enter the market [45]. Considering the diversity of factors affecting the quality of fruit wine, especially the reciprocal selection between the physicochemical properties of raw materials and fermentation strains [46], further studies will be conducted to reveal the effects of more cultivars, origin, vintage, yeast strains, and fermentation protocol on the quality of strawberry wine to provide more detailed data support for the production of strawberry fruit wine.

## 4. Conclusions

To determine the optimal strawberry cultivars for winemaking, this study examined the physicochemical indicators, functional components, and VOCs of the strawberry juice and wines of four typical strawberry cultivars named ZJ, TCL, BX, and TZ. The study findings revealed that there were significant differences in the physicochemical indexes and content of bioactive compounds among the four strawberry cultivars. A total of 42 VOCs were detected in strawberry juice and wines by HS-SPME-GC-MS, and 3-methyl-1-butanol, linalool, trans-2-pinanol, hexanoic acid, and hexanoic acid ethyl ester were the differential VOCs to identify the strawberry wine samples of different cultivars. Overall, strawberry cultivar ZJ had relatively high VOCs and functional bioactive compounds, indicating that it is more suitable for strawberry wine fermentation. More strawberry cultivars and their heterogeneity in origin need to be evaluated to provide solutions for the quality control of strawberry wine fermentation materials. In addition, we will combine the sensory evaluation of strawberry wines from different cultivars in a follow-up study to analyze sensory quality more comprehensively.

## Figures and Tables

**Figure 1 foods-13-02874-f001:**
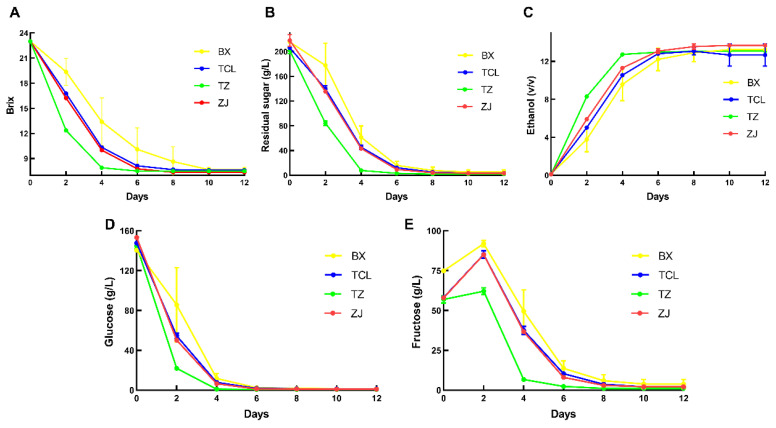
Fermentation kinetics analysis of different cultivars of strawberry wine. (**A**) Brix; (**B**) residual sugar content; (**C**) ethanol content; (**D**) glucose content; (**E**) fructose content; BX: Snow White; TCL: Sweet Charlie; TZ: Tongzhougongzhu; ZJ: Akihime.

**Figure 2 foods-13-02874-f002:**
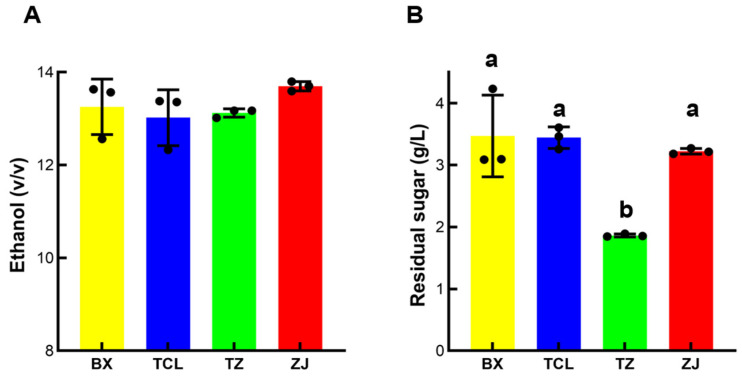
Ethanol and residual sugar content of different cultivars of strawberry wine. (**A**) Ethanol content; (**B**) residual sugar content; BX: Snow White; TCL: Sweet Charlie; TZ: Tongzhougongzhu; ZJ: Akihime. Means with the same letter are not significantly different from each other (*p* < 0.05).

**Figure 3 foods-13-02874-f003:**
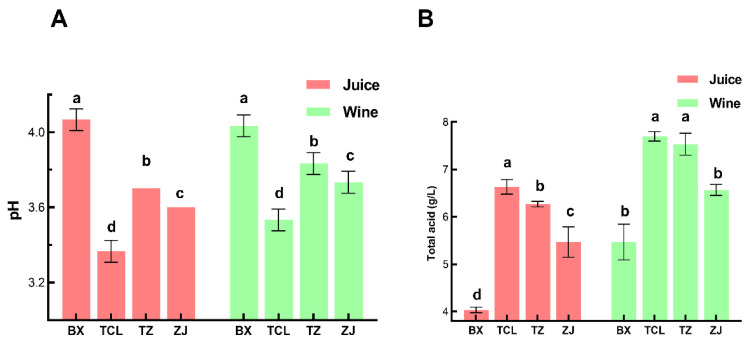
Total acid content and pH of different cultivars of strawberry juice and wines. (**A**) pH; (**B**) total acid content; BX: Snow White; TCL: Sweet Charlie; TZ: Tongzhougongzhu; ZJ: Akihime. Means with the same letter are not significantly different from each other (*p* < 0.05).

**Figure 4 foods-13-02874-f004:**
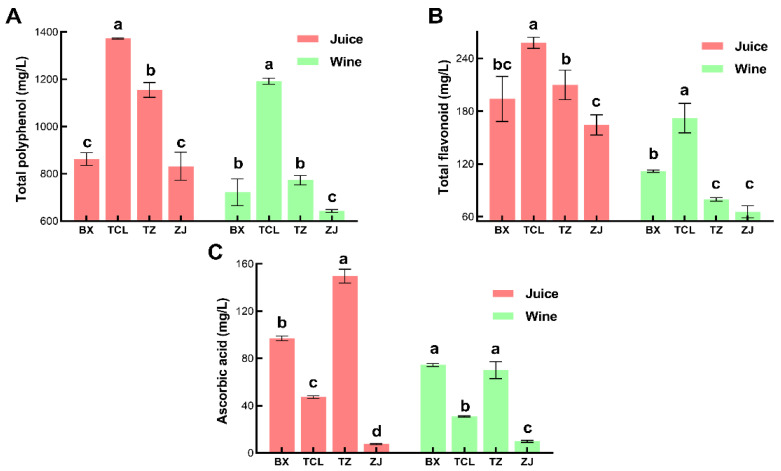
Total polyphenol, total flavonoid, and ascorbic acid content of different cultivars of strawberry juice and wines. (**A**) Total polyphenol content; (**B**) total flavonoid content; (**C**) ascorbic acid content; BX: Snow White; TCL: Sweet Charlie; TZ: Tongzhougongzhu; ZJ: Akihime. Means with the same letter are not significantly different from each other (*p* < 0.05).

**Figure 5 foods-13-02874-f005:**
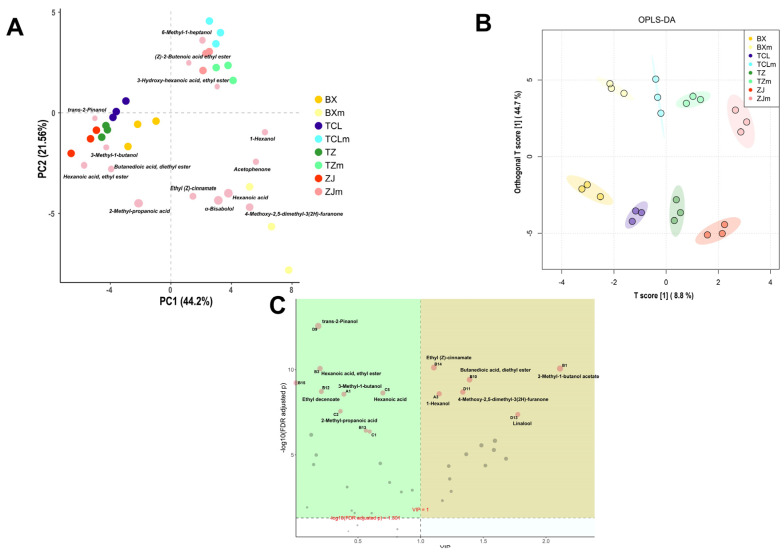
Coordinate analysis and variable importance in the projection (VIP) analysis of VOCs in strawberry juice and wines of different cultivars. (**A**) PCA; (**B**) OPLS-DA; (**C**) VIP value. BX: Snow White; TCL: Sweet Charlie; TZ: Tongzhougongzhu; ZJ: Akihime. Juice (m) samples, BXm: Snow White; TCLm: Sweet Charlie; TZm: Tongzhougongzhu; ZJm: Akihime.

**Figure 6 foods-13-02874-f006:**
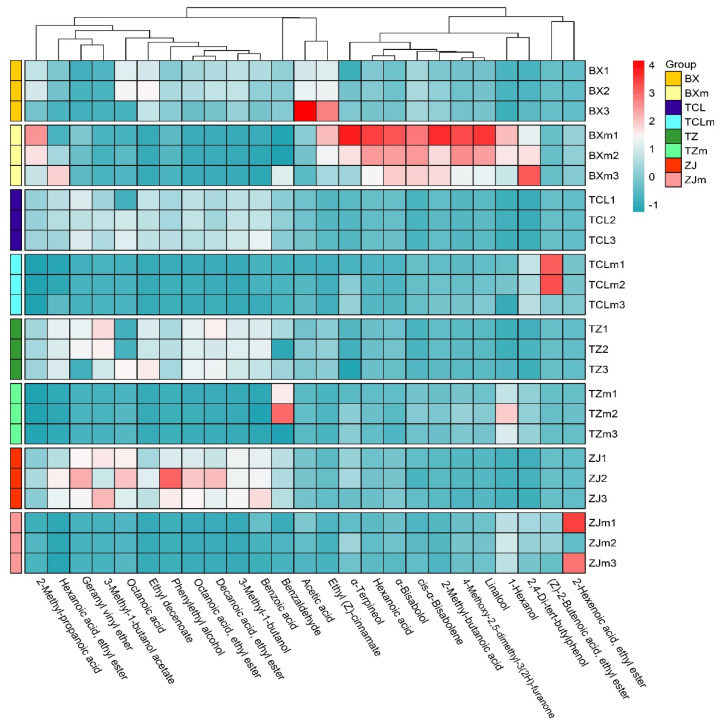
Cluster heatmap analysis based on differential VOCs in strawberry juice and wine samples. Wine samples, BX: Snow White; TCL: Sweet Charlie; TZ: Tongzhougongzhu; ZJ: Akihime. Juice (m) samples, BXm: Snow White; TCLm: Sweet Charlie; TZm: Tongzhougongzhu; ZJm: Akihime.

**Table 1 foods-13-02874-t001:** L* (A), a* (B), and b* (C) values of different cultivars of strawberry wine.

Cultivar	L*	a*	b*	E
BX	17.73 ± 0.62 ^b^	5.71 ± 0.15 ^c^	0.01 ± 0.05 ^b^	19.77 ± 0.47 ^b^
TCL	15.32 ± 0.21 ^c^	8.37 ± 0.19 ^a^	0.45 ± 0.04 ^a^	16.48 ± 0.29 ^c^
TZ	20.62 ± 0.42 ^a^	4.91 ± 0.42 ^d^	0.44 ± 0.03 ^a^	21.20 ± 0.32 ^a^
ZJ	17.41 ± 0.55 ^b^	6.91 ± 0.19 ^b^	0.50 ± 0.11 ^a^	18.95 ± 0.79 ^b^

Mean ± standard deviation (*n* = 3) followed by different letters within each row indicate significant differences (Duncan test, 5%). BX: Snow White; TCL: Sweet Charlie; TZ: Tongzhougongzhu; ZJ: Akihime.

**Table 2 foods-13-02874-t002:** Concentrations of different VOCs in different cultivars of strawberry juice and wines.

Compounds (μg/L)	Strawberry Wine				Strawberry Juice			
	ZJ	TCL	TZ	BX	ZJ	TCL	TZ	BX
Alcohols	8840.42 ± 262.51 ^a^	6740.08 ± 435.05 ^c^	7616.55 ± 315.01 ^b^	5861.23 ± 1158.84 ^d^	385.66 ± 18.03 ^e^	158.3 ± 49.02 ^e^	412.86 ± 104.44 ^e^	474.81 ± 118.12 ^e^
Esters	3788.21 ± 522.52 ^a^	2572.42 ± 114.46 ^b^	3702.41 ± 95.95 ^a^	2372.16 ± 284.91 ^b^	648.66 ± 326.82 ^d^	1347.72 ± 721.15 ^c^	172.61 ± 9.41 ^d^	1362.22 ± 94.01 ^c^
Acids	1264.55 ± 151.9 ^bc^	923.47 ± 260.59 ^cd^	898.34 ± 299.29 ^cd^	1916.27 ± 146.13 ^b^	495.64 ± 66.09 ^cd^	398.27 ± 108.46 ^d^	650.04 ± 121.3 ^cd^	3571.82 ± 1042.61 ^a^
Terpene derivatives	1152.78 ± 116.22 ^b^	874.52 ± 17.81 ^b^	813.02 ± 300.74 ^b^	815.71 ± 29.38 ^b^	700.76 ± 149.22 ^b^	607.76 ± 168.5 ^b^	987.23 ± 247.32 ^b^	3379.7 ± 961.22 ^a^
Phenyl compounds	1628.16 ± 541.18 ^a^	901.06 ± 80.29 ^b^	919.86 ± 64.43 ^b^	714.39 ± 122.59 ^b^	156.47 ± 17.69 ^c^	162.61 ± 4.51 ^c^	141.28 ± 3 ^c^	344.61 ± 31.4 ^c^
Others	71.43 ± 4.4 ^b^	100.96 ± 19.4 ^b^	97.76 ± 7.57 ^b^	215.87 ± 5.41 ^b^	186.51 ± 8.98 ^b^	161.29 ± 60.14 ^b^	189.04 ± 45.61 ^b^	1070.14 ± 378.52 ^a^

Mean ± standard deviation (*n* = 3) followed by different letters within each row indicate significant differences (Duncan test, 5%). BX: Snow White; TCL: Sweet Charlie; TZ: Tongzhougongzhu; ZJ: Akihime. All volatile compounds were identified by comparing the mass spectrometry data with the NIST20 library. 4-methyl-2-pentanol (50 mg/L) was utilized as internal standard and the quantitative analysis of identified volatile compounds was performed by comparing the peak areas with the internal standard.

## Data Availability

The original contributions presented in the study are included in the article and Appendix A, further inquiries can be directed to the corresponding author.

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
