# Peer review of "Effects of Cultivar Factors on Fermentation Characteristics and Volatile Organic Components of Strawberry Wine"

_foods, 2024, doi:10.3390/foods13182874_

Round 1

Reviewer 1 Report

Comments and Suggestions for Authors

In this work, the authors discuss the Effects of Cultivar Factors on Fermentation Characteristics and Volatile Organic Components of Strawberry Wine. The topic is interesting and the manuscript has a potential from a scientific point of view. However, there are a few points that require the authors’ attention.

Introduction is in general well-written. Only a few points should be improved:

1.      Lines 50-51: The authors should provide a few examples of studies examining different fruit wines (e.g. https://doi.org/10.3390/antiox10060889; https://doi.org/10.3390/foods11223619; etc)

2.      Lines 65-86: No need for so detailed explanations at this point. In fact, some of this information could be used in the discussion section.

3.      Line 70: The aim paragraph should be more precise. Focus on the aim objective of this work, which is strawberry wine.

Materials and methods is well organized:

4.      The lack of sensory evaluation for the fruit wines is however disappointing. I believe this is crucial for this kind of work.

Results & Discussion:

5.      In subsection 3.1 the authors say: “It is a trend to improve the aroma of fruit wine and reduce the alcohol content through the mixed fermentation applying non-saccharomyces cerevisiae”. Firstly, a proper reference should be added to this statement. However, the authors should also keep in mind that lowering alcohol content and enhancing aroma can be achieved through other strategies as well: for example the use of mixed cultures that utilize sugars simultaneously (https://doi.org/10.1002/jsfa.10363) or even exploiting other physiochemical factors like high temperature (https://doi.org/10.3390/fermentation7020045)

6.      Despite having some interesting results, the whole “Results & Discussion” section is mostly oriented to describe results and no real discussion is provided. The discussion should be better organized  and expanded so that the authors’ findings (and their importance) could be linked to existing literature. The authors are encouraged to compare their results to the fermentation of other strawbbery cultivars or even similar works examining fermentation with different fruit cultivars.

7.      The authors keep referring to the importance of their results in correlation to the sensory characteristics. However, the lack of sensory trial results is a clear disadvantage for this work. In order to overcome this fact, I recommend the authors create a final paragraph in the discussion section that should contain and highlight all important aspects about the methodology proposed in the manuscript. The authors should expand to the technological, market/economical, environmental, nutritional, etc insights gained from this work.

8.      Conclusions are ok, but any plans regarding the continuation of this research should be added here, as a closing statement.

Comments on the Quality of English Language

Minor editing

Author Response

Detailed response to the reviewers regarding our submission foods-3157143

We appreciate for the honorable editors and reviewers for your valuable feedback. As per suggestions of reviewers, we have made substantial modifications to our manuscript. We also proofread the manuscript to correct typographical, grammatical, and bibliographical errors. In the revised version, changes to our manuscript were all marked red. Please see below, in blue, for a point-by-point response to the reviewers’ comments and concerns.

Thank you again for your positive comments and valuable suggestions to improve the quality of our manuscript.

Responds to the reviewers’ comments:

Reviewer #1

Comment 1. Lines 50-51: The authors should provide a few examples of studies examining different fruit wines (e.g. https://doi.org/10.3390/antiox10060889; https://doi.org/10.3390/foods11223619; etc)

Reply: Thank you very much for your valuable suggestions, relevant research examples of different fruit wines have been added to the revised manuscript (Lines 52-53).

Comment 2. Lines 65-86: No need for so detailed explanations at this point. In fact, some of this information could be used in the discussion section.

Reply: Thanks very much for the precious advice and the relevant content has been reorganized into the discussion section in the revised manuscript.

Comment 3. Line 70: The aim paragraph should be more precise. Focus on the aim objective of this work, which is strawberry wine.

Reply: Thanks very much for the valuable advice, we fully agree with your opinions, the paragraph has removed redundant content to focus on the aim objective strawberry wine (Lines 66-82).

Comment 4. The lack of sensory evaluation for the fruit wines is however disappointing. I believe this is crucial for this kind of work.

Reply: Thanks very much for your valuable advice, we fully agree with your opinions, however, the main objective of this study is to quantify the differences in physicochemical indexes and volatile components of different strawberry varieties, we will combine sensory evaluation in the follow-up study to analyze sensory quality more comprehensively.

Comment 5. In subsection 3.1 the authors say: “It is a trend to improve the aroma of fruit wine and reduce the alcohol content through the mixed fermentation applying non-saccharomyces cerevisiae”. Firstly, a proper reference should be added to this statement. However, the authors should also keep in mind that lowering alcohol content and enhancing aroma can be achieved through other strategies as well: for example the use of mixed cultures that utilize sugars simultaneously (https://doi.org/10.1002/jsfa.10363) or even exploiting other physiochemical factors like high temperature (https://doi.org/10.3390/fermentation7020045).

Reply: Thanks very much for the valuable advice, references have been added to this section, and relevant studies on reducing alcohol content of fruit wines have been added (Lines 180-182).

Comment 6. Despite having some interesting results, the whole “Results & Discussion” section is mostly oriented to describe results and no real discussion is provided. The discussion should be better organized and expanded so that the authors’ findings (and their importance) could be linked to existing literature. The authors are encouraged to compare their results to the fermentation of other strawbbery cultivars or even similar works examining fermentation with different fruit cultivars.

Reply: Thank you very much for your suggestion, references have been added to this section, the discussion section in the revised manuscript has been rewritten to increase its depth and highlight the research significance of this paper, exploring follow-up research ideas.

Comment 7. The authors keep referring to the importance of their results in correlation to the sensory characteristics. However, the lack of sensory trial results is a clear disadvantage for this work. In order to overcome this fact, I recommend the authors create a final paragraph in the discussion section that should contain and highlight all important aspects about the methodology proposed in the manuscript. The authors should expand to the technological, market/economical, environmental, nutritional, etc insights gained from this work.

Reply: Thank you very much for your reminder and suggestion, the last paragraph of the revised manuscript has been rewritten to comprehensively evaluate the research conclusions and practical significance of this manuscript.

Comment 8. Conclusions are ok, but any plans regarding the continuation of this research should be added here, as a closing statement.

Reply: Thank you very much for your reminder and suggestion, the follow-up research plan has been added to the conclusion section (lines 335-337).

Reviewer 2 Report

Comments and Suggestions for Authors

Manucript ID: foods-3157143

Title: Effects of Cultivar Factors on Fermentation Characteristics and Volatile Organic Components of Strawberry Wine

The study explores the differences between strawberry juices and wines obtained from four cultivars of strawberries. For that purpose, strawberry juices and corresponding wines were characterized in terms of physicochemical indicators (total sugar content, pH, total acid, color etc.), bioactive compounds (total polyphenols, flavonoids, vitamin C), and volatile organic components. Study findings indicate potential for preservation of strawberry fruit through the production of functional beverages, thus encompassing short shelf life of this very much appreciated fruit. Furthermore, the findings provide a basis for the selection of strawberries cultivars with superior fermentation characteristics.

According to the above mentioned study goals, the work fits the journal scope. Despite the multitude of methods applied for the characterization of fruit and fruit wine, the manuscript is written in an appropriate way and presented in a well-structured manner. The study is correctly designed and technically sound, relevant for the field. The methods are mostly described with sufficient details. The exception is HS-SPME-GC-MS, a state-of-the-art methods, but some very important information is lacking from the method description   (see comment below). The statistical analyses and statistical reporting are appropriate and well described. The results are significant and interpreted appropriately and consistently throughout the manuscript, providing an advancement of the current knowledge. However, a broader context should be provided based on the data from relevant published studies. The tables and figures are appropriate, easy to interpret and understand, but in case of figure 5, the readability should be improved. Cited references are mostly very recent publications and relevant. Data supporting the findings of the study are available within the manuscript. The conclusions are justified and consistent with the evidence and arguments presented.

Comments:

Line 23 and Line 205: anthocyanins are only mentioned in the abstract (line 23) and the title of section 3.3. However, considering that no methods and no results were presented related to anthocyanins, they should be removed from the mentioned lines.

Line 39: please provide additional information related to mentioned - 5 billion every year – (money, in which currency).

Line 43: although it is understandable, this sentence should be better formulated.

Line 96: as this is their first mentioning in the main text of the manuscript, full names of cultivars should be given: Akihime (ZJ), Sweet Charlie (TCL), 17 Snow White (BX) and Tongzhougongzhu (TZ)

Lines 104-105: numerical value at the beginning of the sentence should be given as text. Use SI units, as in the remaining part of the manuscript, not ppm. SO2  - 2 should be in subscript. Add space between the number and unit in 300mg.

Line 142: chromatographic column used for the analysis should be identified in the manuscript.

Line 162: in Results and Discussion, only some general comments are related to literature references. Authors should provide relevant context for their research considering other published studies (e.g. references number 8 and 13 are mentioned only in the introduction part of the manuscript). Even if there are no references to strawberries, suitable comments can be made for other types of fruit.

Line 205: in the title of section 3.3., anthocyanins should be substituted with Vitamin C.

Figure 2, Figure 3, Figure 5: figures should be positioned after their first mentioning in the text, not immediately after the section title

Figure 4: refer to the fig 4 in the preceding text

Lines 227-8: 70 is not above 70, it should be maintained at or above 70

Table 2: tables should be positioned after the first mentioning in the text, not immediately after the section title

Table 2 and Table S1: Information that the quantitative analysis of identified volatile compounds was performed by comparing the peak areas with the internal standard as well as identification of the internal standard must be provided in the footnotes of these tables.

Figure 5: the readability of the figure should be improved

Author Response

Detailed response to the reviewers regarding our submission foods-3157143

We appreciate for the honorable editors and reviewers for your valuable feedback. As per suggestions of reviewers, we have made substantial modifications to our manuscript. We also proofread the manuscript to correct typographical, grammatical, and bibliographical errors. In the revised version, changes to our manuscript were all marked red. Please see below, in blue, for a point-by-point response to the reviewers’ comments and concerns.

Thank you again for your positive comments and valuable suggestions to improve the quality of our manuscript.

Responds to the reviewers’ comments:

Reviewer #2

Comment 1. Line 23 and Line 205: anthocyanins are only mentioned in the abstract (line 23) and the title of section 3.3. However, considering that no methods and no results were presented related to anthocyanins, they should be removed from the mentioned lines.

Reply: Apologize for the carelessness, Thanks very much for your reminder and the relevant information has been corrected in the revised manuscript (line 23 and line 202).

Comment 2. Line 39: please provide additional information related to mentioned - 5 billion every year – (money, in which currency).

Reply: Thank you very much for your reminder and suggestion, and the relevant information has been corrected in the revised manuscript (line 40).

Comment 3. Line 43: although it is understandable, this sentence should be better formulated.

Reply: Thank you very much for your reminder and suggestion, and the relevant information has been corrected in the revised manuscript (line 43-45).

Comment 4. Line 96: as this is their first mentioning in the main text of the manuscript, full names of cultivars should be given: Akihime (ZJ), Sweet Charlie (TCL), 17 Snow White (BX) and Tongzhougongzhu (TZ)

Reply: Thank you very much for your valuable suggestion. Modification have been conducted in the revised manuscript (line 95).

Comment 5. Lines 104-105: numerical value at the beginning of the sentence should be given as text. Use SI units, as in the remaining part of the manuscript, not ppm. SO2  - 2 should be in subscript. Add space between the number and unit in 300 mg.

Reply: Thank you very much for your kind reminder and modification have been conducted in the revised manuscript (line 103-104).

Comment 6. Line 142: chromatographic column used for the analysis should be identified in the manuscript.

Reply: Thank you very much for your kind reminder and the column information has been added to the revised manuscript (line 142)

Comment 7. Line 162: in Results and Discussion, only some general comments are related to literature references. Authors should provide relevant context for their research considering other published studies (e.g. references number 8 and 13 are mentioned only in the introduction part of the manuscript). Even if there are no references to strawberries, suitable comments can be made for other types of fruit.

Reply: Thanks very much for the valuable advice, related references has been added for an extended discussion on this section (Lines 180-182).

Comment 8. Line 205: in the title of section 3.3., anthocyanins should be substituted with Vitamin C.

Reply: Thank you very much for your reminder, and the relevant information has been corrected in the revised manuscript (line 208).

Comment 9. Figure 2, Figure 3, Figure 5: figures should be positioned after their first mentioning in the text, not immediately after the section title

Reply: Thank you very much for your reminder and suggestion, the corresponding figures have been reformatted as required in the revised manuscript.

Comment 10. Figure 4: refer to the fig 4 in the preceding text

Reply: Thank you very much for your reminder and modification have been conducted in the revised manuscript.

Comment 11. Lines 227-8: 70 is not above 70, it should be maintained at or above 70

Reply: Thank you very much for your kind reminder and modification have been conducted in the revised manuscript (line 227).

Comment 12. Table 2: tables should be positioned after the first mentioning in the text, not immediately after the section title.

Reply: Thank you very much for your reminder and suggestion, Table 2 has been reformatted as required in the revised manuscript.

Comment 13. Table 2 and Table S1: Information that the quantitative analysis of identified volatile compounds was performed by comparing the peak areas with the internal standard as well as identification of the internal standard must be provided in the footnotes of these tables.

Reply: Thank you very much for your kind reminder and modification have been conducted in the revised manuscript.

Comment 14. Figure 5: the readability of the figure should be improved

Reply: Thank you very much for your kind reminder, we have modified the annotations to increase the readability for Figure 5.

Round 2

Reviewer 1 Report

Comments and Suggestions for Authors

The authors have addressed the majority of my comments and the manuscript has been substantially improved. I would prefer the authors to ideally compare their results to the fermentation of other strawbbery cultivars or even similar works examining fermentation with different fruit cultivars, but I understand that this information may not be yet available in literature.

However, I still believe the authors should expand the final paragraph of the discussion section (lines 304-317) and should add some more information about all important aspects gained from this work (technological, market/economical, environmental, nutritional, etc). 

The lack of sensory evaluation is important. In order to be circumvent it, sensory evaluation should be referred as future plan at the last line of conclusion paragraph. Otherwise this work seems only half of a whole.

Comments on the Quality of English Language

Minor editing

Author Response

We appreciate for the honorable editors and reviewers for your valuable feedback. Following your suggestions, we have rewritten the discussion section to increase the depth and highlighted the economic significance of this study for the strawberry industry, as well as subsequent research plan. We also proofread the manuscript to correct typographical, grammatical, and bibliographical errors. In the revised version, changes to our manuscript were all marked red.

Thank you again for your positive comments and valuable suggestions to improve the quality of our manuscript.

Reviewer #1 (Round 2

Comment 1. The authors have addressed the majority of my comments and the manuscript has been substantially improved. I would prefer the authors to ideally compare their results to the fermentation of other strawbbery cultivars or even similar works examining fermentation with different fruit cultivars, but I understand that this information may not be yet available in literature. However, I still believe the authors should expand the final paragraph of the discussion section (lines 304-317) and should add some more information about all important aspects gained from this work (technological, market/economical, environmental, nutritional, etc). 

Reply: Thank you very much for your valuable suggestions. It is our negligence that we failed to describe clearly, and the relevant research examples of different fruit wines have been added to the revised manuscript (Lines 325-339), and added some references, as following.

34、Čakar, U.; Petrović, A.; Pejin, B.; Čakar, M.; Živković, M.; Vajs, V.; Đorđević, B. Fruit as a substrate for a wine: A case study of selected berry and drupe fruit wines. Scientia Horticulturae. 2019, 244, 42-49. https://doi.org/10.1016/j.scienta.2018.09.020

35、Fernandes, A.; Oliveira, J.; Teixeira, N.; Mateus, N.; De Freitas, V. A review of the current knowledge of red wine colour. Oeno One. 2017, 51, 1-15.

36、Sun, X.; Yan, Z.; Zhu, J.; Wang, Y.; Li, B.; Meng, X. Effects on the color, taste, and anthocyanins stability of blueberry wine by different contents of mannoprotein. Food Chemistry. 2019, 279, 63-69. https://doi.org/10.1016/j.foodchem.2018.11.139

Comment 2. The lack of sensory evaluation is important. In order to be circumvent it, sensory evaluation should be referred as future plan at the last line of conclusion paragraph. Otherwise this work seems only half of a whole.

Reply: Thanks very much for the precious advice and the relevant content has been reorganized into the discussion section in the revised manuscript (Lines 325-339). Follow your comments, we added some contents at the last line of conclusion paragraph, as “ In addition, we will combine sensory evaluation of different cultivars strawberry wines in the follow-up study to analyze sensory quality more comprehensively (Lines 353-355).”
